# Point-of-care diagnostic technology in paediatric ambulatory care: a qualitative interview study of English clinicians and stakeholders

Meriel Elizabeth Raymond  ,[1] Chris Bird,[1,2] Oliver van Hecke,[1,3] Margaret Glogowska,[1,3] Gail Hayward  [1,3]

[1]Nuffield Department of Primary Care Health Sciences, University of Oxford, Oxford, Oxfordshire, UK
[2]Emergency Department, Birmingham Children's Hospital, Birmingham, UK
[3]Nuffield Department of Primary Care Health Sciences, Oxford University, Oxford, UK

**Correspondence to**
Dr Meriel Elizabeth Raymond;
meriel.raymond@phc.ox.ac.uk

## ABSTRACT

Point-of-care (POC) tests have the potential to improve paediatric healthcare. However, both the development and evaluation of POC technology have almost solely been focused on adults. We aimed to explore frontline clinicians' and stakeholders' current experience of POC diagnostic technology in children in England; and to identify areas of unmet need.

**Design, setting and participants** Qualitative semistructured telephone interviews were carried out with purposively sampled participants from clinical paediatric ambulatory care and charity, industry and policymaking stakeholders. The interviews were audio-recorded, transcribed and analysed thematically.

**Results** We interviewed 19 clinicians and 8 stakeholders. The main perceived benefits of POC tests and technologies were that they aided early decision-making and could be convenient and empowering when used independently by patients and families. Clinicians and stakeholders wanted more POC tests to be available for use in clinical practice. Most recognised that play and reward are important components of successful POC tests for children. Clinicians wanted tests to give them answers, which would result in a change in their clinical management. Detecting acute serious illness, notably distinguishing viral and bacterial infection, was perceived to be an area where tests could add value. POC tests were thought to be particularly useful for children presenting atypically, where diagnosis was more challenging, such as those less able to communicate, and for rare serious diseases. Many participants felt they could be useful in managing chronic disease.

**Conclusions** This exploratory study found that clinicians and stakeholders supported the use of diagnostic POC technology in paediatric ambulatory care settings in England. Some existing tests are not fit for purpose and could be refined. Industry should be encouraged to develop new child-friendly tests tackling areas of unmet need, guided by the preferred characteristics of those working on the ground.

## STRENGTHS AND LIMITATIONS OF THIS STUDY

⇒ Semistructured interviews enabled an in-depth exploration of the experiences of the heterogeneous participants with different backgrounds.
⇒ Purposive sampling with snowballing facilitated the interviewing of a broad range of clinicians and stakeholders on this topic. Inclusion of stakeholders enabled emergence of views from policymaking and industry perspectives.
⇒ However, the broad remit of the study meant that we were unable to cover every single test and paediatric clinical presentation, making 'data saturation' difficult to achieve.
⇒ Although children's and parents' perspectives were mentioned by our participants, and some offered their own experiences as parents, their views were not specifically sought in this study.
⇒ All participants were based in England. As such, our findings are applicable to English stakeholders and clinicians and may not be transferable to other settings.

## INTRODUCTION

Paediatric ambulatory care places huge demand on healthcare services. One in four consultations in ambulatory care in the UK are for children.[1][2] Children present with a different disease spectrum to adults, having a high incidence of acute infections.[3] Most of these consultations are for upper respiratory tract infections, which are generally self-limiting. The incidence of serious infection in children presenting to primary care has been estimated to be less than 1%.[3] The challenge in primary care is that these serious infections often present with non-specific symptoms, especially in the early stages. Furthermore, children have the potential to deteriorate more quickly than adults.[4] It is difficult to detect those children who will progress to serious illness requiring secondary care input in a timely way.[5] Inappropriate prescribing, unnecessary referrals to hospital and needless additional testing often result from this diagnostic uncertainty.[6] There was a 10%–20% trend increase in potentially avoidable, short stay hospital admissions of children in

England from 1997 to 2012.[7–9] The onset of the SARS-CoV-2 pandemic in March–April 2020 saw 69% less children attending emergency departments in the UK[10]; this was followed by a 1%–4% increase in attendance per week. Paediatric emergency research groups have identified the need to develop better diagnostics for 'low numbers, high stakes diagnoses' in children.[11–13]

Point-of-care (POC) tests can be defined as any test performed near a patient or clinic with results available during a clinical visit.[14 15] POC technology includes measurements taken at the bedside, such as smartphone applications and wearables. POC tests have the potential to reduce diagnostic uncertainty in acute illness and streamline management of chronic disease, improving clinical outcomes and reducing health-related costs.[5] A systematic review and meta-analysis of the clinical impact of POC tests in paediatric ambulatory care found few studies.[5] The use of malarial POC tests was found to reduce overtreatment by a third compared with usual care. HIV–POC tests improved early initiation of antiretroviral therapy compared with usual care. POC C reactive protein may reduce immediate antibiotic prescribing for respiratory tract infections in low-income and middle-income countries, but evidence was lacking in high-income countries. The evaluation of POC tests for children often lags behind that for adults, for example, with SARS-CoV-2 testing.[16]

Attitudes of primary care clinicians towards POC blood tests in Europe and Australia have been synthesised in one systematic review of qualitative studies.[14] Participants thought that POC testing improved diagnostic certainty, treatment, self-management of chronic disease, clinician–patient relationships and perceived patient experience. The views of English paediatricians and emergency department healthcare providers on the use of POC tests to assess febrile children have also been explored.[17] This study agreed with previous publications on POC tests' advantages—improved patient flow, quicker decision-making, minimal invasiveness of testing and improved antibiotic stewardship—but also had concerns about a decrease in clinical acumen, the reliability of POC tests and the issue that some POC tests with a continuous variable made clinical decision-making more, not less, difficult. This paper suggested seeking the views of paediatricians in district general hospitals, general practitioners (GPs) and other paediatric subspecialities.

Other recent studies have highlighted obstacles to greater use of POC tests in children. Pandey et al, in a survey of UK children's emergency departments and paediatric assessment units, found lack of funding, a lack of evidence and governance issues surrounding quality assurance of tests, meant several new biomarkers, which already exist, had not been adopted in the majority of units.[18] Rasti et al, in a qualitative survey of nurses and doctors in a Swedish children's emergency department, found that while POC tests benefits included better satisfaction from families who wanted a test for their child and greater reassurance in some instances in clinical decision-making, those surveyed feared the use of POC tests in hospital and at home might drive more unnecessary testing and that reliance on POC tests could diminish clinical skills.[19]

Little is known about attitudes of primary care clinicians towards POC tests in children other than blood tests,. There is little information on stakeholders' views or views towards POC technologies, including apps and wearables.

The diagnostic needs in paediatric ambulatory care are unlikely to be met by diagnostics which have been developed with an adult population primarily in mind. Children are not 'mini adults' and have specific needs that should be addressed in order for diagnostics to be helpful in a clinical setting. These might include the requirement for rapid diagnosis, smaller sample volumes and less invasive procedures. POC tests have the potential to address these needs. In order to stimulate the development and evaluation of POC diagnostic technology which is of the greatest benefit in paediatric healthcare, it is important to understand the current experience of those using these technologies and identify areas of unmet need. We aimed to seek the views and experiences of a broad range of clinicians and stakeholders with an interest in paediatric ambulatory care in the UK about current usage and unmet needs for POC diagnostic technology.

## METHODS

Qualitative research is highly appropriate for capturing and exploring people's experiences and perceptions; and has considerable power to explain actions, decisions and processes.[20] Therefore, qualitative interviews were used to explore perceptions of clinicians and stakeholders towards POC tests and technologies in paediatric ambulatory care.

### Sampling and recruitment

A maximum variation, purposive sample of participants was sought based on gender, level of clinical experience and range of National Health Service (NHS) settings.[21] We advertised for participants using the Paediatric Emergency Research in the UK and Ireland mailing list in August 2019 and April 2020, and on the website for the Nuffield Department of Primary Care Health Sciences, University of Oxford, at www.phc.ox.ac.uk/iTAP, from 19 June 2019.

We directly approached specialist clinicians, children's commissioners, clinical commissioning groups (CCGs; groups of general practices which come together in each area to commission services for their patients and population), children's charities pertaining to serious illness and Technology Innovation Transforming Child Health using telephone or email details that were in the public domain.

Recruitment was extended to contacts of participants in a 'snowballing' effect. Early interviews shaped the identification of further interviewees, using a principle

of grounded theory; namely, theoretical sampling which permits the deliberate inclusion of participants whose viewpoints have been shown to be of interest.[22] The decision to stop interviewing, when sufficient information had emerged and there was satisfactory explanation for the emerging themes, was discussed and agreed among the research team.

## Interviews

Qualitative semistructured individual interviews were conducted by the primary researcher, MER. These enabled in-depth exploration of the experiences of the heterogeneous participants,[23] through interviewer and interviewee interaction, and exploration of details, which were significant to either party as the interview progressed. A focus group discussion of a wide range of professionals would be less likely to capture these individual experiences. Focus-group discussion was also avoided due to logistical difficulty in arranging group clinician sessions, need for Health Research Authority approval for interviews occurring on NHS premises and divergence of stakeholder interests.

Participants were offered a telephone or face-to-face interview of around 30 min. Due to participant preference and the COVID-19 pandemic, all interviews were conducted by telephone. Informed verbal consent was obtained prior to interview. Draft topic guides for the interviews with clinicians and stakeholders were developed to address the study objectives (see online supplemental material 1). These were based on the available literature, and drew on issues from topic guides for other studies we have conducted around clinicians' views of POC testing.[24 25] The topic guide was initially reviewed by the research team, modified iteratively by the primary researcher based on feedback and amended after 12 interviews following discussion with the research team. Participants were informed 'by POC tests and technologies, we mean any diagnostic technology to include tests on bodily fluids, imaging, wearables, digital technology and smartphone apps'. Interviews were recorded using a digital audio recorder and transcribed verbatim by a single professional transcriber. Field notes were made by the primary researcher during and after the interviews. Data were stored and processed in line with General Data Protection Regulation. In recognition of the time contributed to the study, interviewed participants were offered a £20 gift voucher.

## Analysis

Transcripts were anonymised and checked against the audio recordings for accuracy. Anonymised transcripts were uploaded into a specialist software programme to assist organisation of data (NVivo V.12). A 'ground up' approach from the data was adopted to analyse the complete data set[26] using thematic analysis.[23] The primary researcher read and familiarised herself with the transcripts. Systematic and detailed codes were compared and grouped to create categories. These were organised into an initial 'data driven' coding framework based on six coded interviews. These interviews were read by MG and GH and the coding framework checked. This coding framework was iteratively applied to subsequent transcripts. 'Constant comparison' was used to cross-check ideas and categories that were emerging across interviews, taking an inductive approach.[20] Broad themes were developed using 'single sheet' brainstorming.[20] Agreement on coding, themes and subthemes was sought between members of the research team. An audit trail from the raw data of the interview transcripts through coding to development of themes was established to ensure dependability. Participants were provided with the results section and given 2 weeks to provide feedback.

## Researcher characteristics and reflexivity

The primary researcher was a GP undertaking a master's degree in public health. She attended a course on qualitative interviewing prior to the study. The participants were aware of her clinical background prior to interview and her reasons for undertaking the research. MG is a specialist qualitative researcher.

## Public and patient involvement

No patients were involved. The final manuscript was sent to participants.

## RESULTS

Overall, 22 interviews were conducted between June 2019 and July 2020. The interviews lasted an average of 35 min.

## Participant characteristics

For complete participant characteristics please see table 1. Of the 22 participants, 14 were clinicians, 3 stakeholders and 5 were both clinicians and stakeholders.

Of the 19 clinicians, 9 were from primary care (7 GPs, 2 nurses), and 10 from secondary or tertiary care (8 doctors, 2 nurses).

The 8 stakeholders represented three CCGs, three charities and one Tech Company.

## Themes and subthemes

The main themes and subthemes are described below in box 1.

### Theme 1: Potential benefits of POC tests and technologies

#### 1a: POC tests facilitate early decision-making

Participants reported that the predominant advantage of POC tests and technologies is that they give rapid results compared with tests requiring laboratory processing or transfer of the child to another department. They thought that POC tests increased the speed of clinicians' decisions and allowed the assessing clinicians to incorporate the result as part of their holistic assessment. Delayed laboratory results would be more likely to be interpreted by a clinician who had not seen the child.

**Table 1** Complete participant characteristics

| Participant | Job role | Time in that role/years (mo=months) | Gender | Level 1=primary 2=secondary 3=tertiary | Setting Rural=0 Urban=1 | Recruitment 1=PERUKI 2=website 3=direct 4=snowball 5=conference |
|---|---|---|---|---|---|---|
| Clinicians | | | | | | |
| **01** | **Consultant paediatric and neonatal surgeon** **BAPS** | **5** **2** | **M** | **3** **n/a** | **1** | **3** |
| 02 | Consultant paediatrician | 2 | M | 2 | 0 | 1 |
| 03 | GP | 1 mo | F | 1 | 1 | 4 |
| 04 | Specialist asthma nurse | 26 | F | 1 | 1 | 4 |
| **05** | **Macmillan GP** **CCG clinical lead for children and young people** | **12** **5 mo** | **F** | **1** **n/a** | **1** | **3** |
| 06 | Consultant in paediatric and adult emergency medicine | 14 | F | 2 | 1 | 1 |
| 07 | Specialist paediatric trainee | 5 | F | 2 | 1 | 4 |
| 08 | Foundation Year 1 Doctor (junior doctor in their first year of practice) | 2 mo | M | 2 | 1 | 4 |
| **09** | **GP** **CCG clinical lead for cancer, children and maternity** | **20** **8** | **M** | **1** **n/a** | **1** | **2** |
| 10 | GP | 4 | F | 1 | 1 | 5 |
| 11 | Consultant children's orthopaedic surgeon | 4 | M | 3 | 1 | 3 |
| 12 | Primary care advanced nurse practitioner | 35 | F | 1 | 0 | 4 |
| 13 | Consultant community paediatrician | 21 | M | 2 | Mixture | 3 |
| 14 | Consultant community psychiatrist of children and adolescents | 3.5 | F | 2 | Mixture | 3 |
| 15 | Senior staff nurse children's emergency department | 4 | F | 3 | 1 | 3 |
| 16 | Urgent care GP | 18 | M | 1 | 1 | 2 |
| 17 | Primary care advanced nurse practitioner | 23 | F | 1 | 1 | 3 |
| Stakeholders | | | | | | |
| 01 | Meningitis Research Foundation | 2 | F | n/a | n/a | 3 |
| 02 | Little Miracles | 10 | F | n/a | n/a | 3 |
| **03** | **Asthma UK** **GP** | **3** **15** | **M** | **n/a** **1** | **Mixture** | **3** |
| 04 | HappyR health | 1 | M | n/a | n/a | 3 |
| **05** | **CCG clinical lead for children, young people and maternity** **GP** | **20** **7** | **F** | **n/a** **1** | **1** | **3** |

Participants shown in bold are both clinicians and stakeholders.
BAPS, British Association of Paediatric Surgeons; CCG, clinical commissioning group; GP, general practitioner; PERUKI, Paediatric Emergency Research in the UK and Ireland.

**Box 1  Main themes and subthemes**

**Theme 1: Potential benefits of point-of-care (POC) tests and technologies**
1a: POC tests facilitate early decision-making
1b: Home-based POC tests are convenient
1c: POC tests are empowering for children and their families
**Theme 2: Areas for improvement for POC tests and technologies**
2a: POC tests should be more widely available
2b: End-users should find POC tests quick and easy to use
2c: POC tests should be agreeable and engaging for children
2d: POC tests should make a difference to clinical management

you don't really know if this lump is an abscess or not, which can guide your treatment and management; having to rely on a radiologist really delays the treatment of the child and makes you… admit the child for the scan to happen the next day… …if you had the chance to do that by the bedside… that….would really make a difference [Emergency Department Consultant Clinician#6]

A Macmillan GP (GP with palliative care as a specialist interest) thought that availability of POC full blood count in primary care settings would facilitate faster pick-up of difficult-to-diagnose serious conditions such as childhood cancer, as a delay in hospital referral often delayed the diagnosis.

they'd been back and forwards to the GP with tiredness or a bit of a viral infection… and it was only when they got into A&E [Accident and Emergency]… that the blood tests [were] done and the leukaemia was found… probably a barrier for us in primary [care] at the moment is that we would have to refer the patient to… the hospital… but if we could just do it in primary care that probably would… transform that sort of diagnosis. [Macmillan GP, Clinician#5]

Many clinicians and stakeholders thought that POC technologies could help to give earlier diagnosis of chronic disease, enabling prompt appropriate treatment and decreasing morbidity. Examples were given of spirometry and Fractional Exhaled Nitric Oxide (FeNO) (see table 2), POC eosinophils and mental health questionnaires.

Clinicians and stakeholders representing children with additional needs, disabilities and life-limiting conditions, added that early pick-up of clinical deterioration was particularly important, as they often had an *up and down*

**Table 2**  Additional participant quotes listed by theme and subtheme

| Theme | Subtheme | Test/technology | Quote | Participant |
|---|---|---|---|---|
| 1: Potential benefits of POC tests and technologies | 1a: POC tests facilitate early decision-making | Spirometry, FeNO | Tests, such as, spirometry and FeNO are good objective measures which we can use at the bedside to help decide whether… somebody has or doesn't have asthma… a lot of patients get under diagnosed… that means they're getting chronic symptoms and inflammation and ongoing damage within the airways… which can cause… disability from stopping them doing normal things in their life; it can put them at risk of life-threatening asthma attacks and it can cause chronic inflammation of the lungs causing long-term damage. | Stakeholder#3 |
|  | 1b: Home-based POC tests are convenient | Remote observations | from a patient perspective and a practice perspective… seeing as much as we can remotely is… much better. Nobody in their right mind wants to bring a sick child out and sit in a doctor's surgery waiting for a doctor or practitioner to be running late [when] the kid's not well | Clinician#17 |
| 2: Areas for improvement for POC tests and technologies | 2b: End-users should find POC tests quick and easy to use | Urinalysis | we had an example of a (teenage) girl…with fairly non-specific symptoms… Had not been able to produce the urine, said they would do it later, that didn't happen… the diagnosis was made about perhaps a week later [of] diabetes | Clinician#12 |
|  |  | Smart inhaler | there is one device that clips to one specific inhaler… it measures the sound of the inhalation so you can gauge whether or not… that dose has been taken properly… currently it's only being used in research, but the potential is there | Clinician#4 |
|  |  | Monitoring of exhaled gases | before long there will be the technology that when you talk into your mobile phone it will be able to monitor your asthma… a combined exhaled carbon monoxide and nitric oxide monitor | Clinician#4 |

FeNO, fractional exhaled nitric oxide; POC, point-of-care.

*trajectory and a high risk of sudden episodes of acute illness* [GP Clinician#5]. They thought it might be worth monitoring such children at home to pick up early physiological changes as a *safety net* [GP Clinician#5].

### 1b: Home-based POC tests are convenient
Participants suggested that POC tests performed at home by patients and their families or caregivers could decrease the need for face-to-face assessment in healthcare settings. An example was given of the use of POC clotting testing in children with replacement heart valves *improv[ing] the quality of those families' lives* making a *really big difference* [Community paediatrician Clinician#13]. Participants felt that home testing would be convenient for patients and clinicians and could speed up recognition and escalation of acute illness. Furthermore, it was thought that this would improve infection prevention and control, particularly during the COVID-19 pandemic. An unmet need was identified for the detection of vital signs including temperature and oxygen levels by parents at home, for example, with smartphone cameras (see table 3).

### 1c: POC tests are empowering for children and their families
Participants explained that the additional objective information given by POC tests and technologies to children and their families would empower them to communicate their illness more effectively to healthcare professionals, facilitating the consultation. This was particularly important for the families or carers of children who struggled to communicate because of disability, and in whom detection of illness is more difficult.

> families find communication about a problem with healthcare services quite challenging and if they were equipped with a range of clinical parameters to help their discussion… they might find they access the right kind of healthcare quicker [GP Clinician#5]

Furthermore, participants said that the results from these tests helped children with chronic disease and their families to look after their own health better.

> I have heard of young people using and parents taking control of diabetes management using Apps quite pro-actively…….[they attend] clinic and consultants [feel] a bit redundant because suddenly they've been replaced by this App which is giving their family a lot more control… [they] are actually making those decision themselves about management…we can… empower people to actually self-manage these conditions very effectively [GP Clinician#5]

## Theme 2: Areas for improvement for POC tests and technologies
### 2a: POC tests should be more widely available
Most of the participants had not come across many POC technologies in their clinical practice, or felt that were not widely available. They also thought that cost, for example, of FeNO and peripheral oxygen saturation monitors,

could limit accessibility and lead to *inequitable distribution* [Asthma nurse Clinician#4].

### 2b: End-users should find POC tests quick and easy to use
Many participants felt that POC tests and technologies need to be quick to use, so that a child could be distracted, for instance during a distressing test; or not lose concentration, for instance during measurement of peak flow. The *time-poor* clinicians [GP Clinician#9] also wanted quick tests; first to improve patient flow, and second to enable continuity, in that the same clinician seeing the patient at initial contact could also be responsible for interpreting the result. Some participants expressed a preference for tests that would give results in seconds. Innovations they suggested included contactless scanning to measure oxygen saturations and height (Emergency Department nurse Clinician#15); measurement of basic observations with smartphone cameras (GP Clinician#16) or use of smartphone apps to diagnose rashes (Advanced nurse practitioner Clinician#12).

Participants reported that POC tests need to be easy to perform to avoid causing pain and stress for children and their families. This was particularly true for finger pricks, throat swabs and blood pressure measurements. There was, however, a consensus that finger prick tests using a single drop of blood are acceptable. Many participants stated that urine samples (see table 2), peak flows and spirometry could be challenging for younger children to perform. Participants said that POC tests and technologies requiring no extra effort by the child would be ideal (see table 2, smart inhaler and monitoring of exhaled gases).

Many participants felt that tests and technologies needed to be *fool proof* to perform [Emergency Department Consultant Clinician#6]. Participants reported that where tests were not easy to use, it put them off using them. They frequently gave the example of measuring peripheral oxygen saturations, which posed a logistical challenge in primary care as it was often difficult to obtain a reliable result. One participant stated *there's a gap of a non-single-use [oxygen saturation] probe that is effective and quick to use* [Advanced nurse practitioner Clinician#17].

> With younger kids… under five years of age… and particularly babies under one… we've got one [peripheral oxygen saturations monitor] machine per practice. So first of all, I have to go out and get it, find the box. It might be… in the right place or maybe another clinician's got it. You've got to send a message out, "Who's got the [oxygen saturations] machine?"… it seems to take… four or five minutes sometimes to get a reading. You fidget around, try on the thumb… end up trying earlobes and things… it's just really hard when, on young babies you try across the foot and the kid starts wriggling and kicking… and then if you're unlucky you'll get a bad trace and… it's not actually their sats because the pulse rate's completely wrong… but if it starts to then blip and say things like

**Table 3** Unmet needs: ideas for application of new tests or technologies that have not already been mentioned in table 2

| Test/technology/pathway | Quote | Participant |
|---|---|---|
| **Acute serious illness** | | |
| Predicting severity of bronchiolitis | it's really difficult to tell which, which babies are going to have a mild bronchiolitic course and just settle down quite quickly and those that are going to progress and need additional respiratory support, so… whether there's a breath-activated test… that tells you… [that] would be incredible | Clinician#2 |
| Remote observations using smartphone cameras and apps | we… are wary of sepsis for example… in children who are poorly with acute illnesses we… spend quite a lot of time gaining information about those particular sepsis markers so I will be measuring their respiratory rate. I'll be checking their oxygen levels, I'll be measuring their respiratory rate. I'll be checking their pulse. I'll be checking their blood pressure if that's appropriate. We'll be checking their temperature, their capillary refill time… if a patient could do that [at home] so there is an App which can(quickly and non-invasively) assess these (sepsis) markers… that would be hugely helpful… in making a decision safely…and may mean that less patients need to be assessed face to face or in hospital… it would save us a lot of time and would provide a lot of assistance | Clinician#15 |
| Poisons and seizures | you can send the blood test off and get paracetamol salicylate levels; that's fairly standard… It would be helpful to get those results earlier [with] other drugs… for your older teenager who comes in unconscious and you're wondering what they might have taken… children with epilepsy… are they taking the right dose of sodium valproate?… if you could find that out quickly then would, that would change our management… when they're coming in having a seizure | Clinician#6 |
| Appendicitis | if you had a child who was suspected to have appendicitis clinically, but you wanted to be more certain, then you would have access to… a bedside ultrasound… and prove definitively whether they did or did not… 1) it could provide better selection of children who needed to have treatment for their appendicitis; and 2)… it could give reassurance to those who didn't have appendicitis so they could be sent home | Clinician#1 |
| Ovarian torsion | ultrasound is used for ovarian torsion… [that] could be done at the bedside | Clinician#1 |
| Fracture | avoiding X-rays, doing near patient ultrasound to diagnose your fracture or whatever it is…. some of this stuff can really help with minors, reducing radiation exposure of children and, and speeding up the process | Clinician#6 |
| **Distinguishing bacterial and viral infection** | | |
| Diagnosing bacterial meningitis | you could distinguish viral meningitis and bacterial meningitis to high sensitivity and specificity with this 2 RNA transcript signature | Stakeholder#1 |
| | I have read about the rapid DNA test for Neisseria meningitis… and that will be very useful in the context of a child presenting with non-blanching rash and fever… I tend to over treat these kind of children or to admit for observations waiting for… blood tests to come back | Clinician#7 |
| **Diagnosing and monitoring chronic disease** | | |
| Assessing pain or stress in children unable to communicate | kids with ASD… you could monitor where [and] when their heart rate goes up and when there's more signs of stress, even if they don't realise that they're getting stressed at these times… some objective monitoring could be helpful for those kids because they're not very aware of their own emotions… you can [then] plan an intervention accordingly | Clinician#14 |
| Diagnosing genetic diseases | we're talking of whole genetic sequencing coming along very, very quickly now…getting the results by the bedside | Clinician#13 |

ASD, autistic spectrum disorder; RNA, ribonucleic acid.

80 per cent, you just start thinking, 'Oh God, why the hell did I do this' [GP Clinician#10]

### 2c: POC tests should be agreeable and engaging for children

Many participants felt that POC tests should ideally be enjoyable. The asthma nurse (Clinician#4) described making peak flows into a game. Reward was particularly important in children with disability.

> anything that could be done as a wearable, so that… they're still able to play. A lot of the kids that we have when they go into A&E, they might be really quite poorly but actually… it's usual for them… They just want to be able to play and… get on with their life…. ….and so it's then quite inconvenient and they get upset… and quite angry and quite stroppy… because… it's interfering with their day… anything that we can do to… make it less medicalised and more play-based, more fun [is] always a good thing. [Little Miracles Stakeholder#2]

Visual results such as FeNO were described as engaging the patient and increasing adherence with medication. When children entered information into one stakeholder's app, their progress was indicated by the growth of a plant (Stakeholder#4).

> FeNO is massively useful in patients that are… not adherent with their medication in that it gives them that lightbulb moment to actually visualise what's going on inside the chest… [if] you can then illustrate that by measuring an inflammatory marker, they tend to be a bit more adherent. [Asthma nurse Clinician#4]

### 2d: POC tests should make a difference to clinical management

Participants wanted POC tests and technologies to give them results that would make a difference to their decision-making and get them *further ahead* [Emergency Department Consultant Clinician#6]. They felt that *something objective* [GP Clinician#10] might *stop interpersonal and intrapersonal variance* [Paediatrician Clinician#2]. Many of them expressed a wish for tests with *good sensitivity and specificity to be reliable* [Foundation Year 1 Doctor (junior doctor in their first year of practice) Clinician#8]. Participants wanted confirmatory tests to enable detection of acute serious illness *to rule out the worst-case scenario* [Paediatric trainee Clinician#7]. For instance, many clinicians asserted that low peripheral oxygen saturations would help pick up acute serious illness, and guide referral to hospital, mode of transport to hospital, and need for admission. A GP (Clinician #9) had invested £500 in a machine because of this perceived impact. One participant (Paediatrician Clinician#2) felt that these basic observations were sometimes underutilised in the clinical setting, and that this could be a focus for improvement over the development of new tests or technologies.

> I sometimes don't recognise that people are as bad as they are because I'm a bit too optimistic. But sometimes I'll see a child… and say, 'Actually, you don't look…too bad' and then I'll put the oximetry on and go… 'Oh, actually, you're worse than I realised. Let's just think about this a bit more seriously' [GP Clinician #9]

The acute serious illnesses that participants raised were predominantly sepsis and meningitis, with an emphasis on the need to distinguish between bacterial and viral infection, and confirmation of a specific pathogen being particularly helpful. This could increase clinician confidence in diagnosis and management, including antibiotic prescribing. They gave examples of POC streptococcal PCR and POC respiratory PCR panels in primary care.

> URTI {Upper Respiratory Tract Infection}-type symptoms… the research nurse did [nasopharyngeal swabs] and they could run the analyser and within an hour you would know whether this had a bacterial element to it and then obviously you could prescribe [antibiotics] if that was appropriate… the parents [had] such a willingness to take part in that research trial… the fact that you could say to them, 'Yeah we can test you straight away now,' and we can get an answer to you… parents were very happy with that [Advanced Nurse Practitioner Clinician#12]

The importance of exact pathogen detection in the context of public health was also raised, with implications for contact-tracing and vaccination when meningococci and SARS-CoV-2 were detected. Participants acknowledged that results might offer false reassurance, for example, in a viral respiratory tract infection, and that clinicians would still need to safety net against development of a secondary bacterial infection. Desire for POC tests to assist in diagnosis of non-infective acute serious illness including ischaemia, diabetes, cancer, seizures, poisoning and trauma were also mentioned in the interviews; as were tests to diagnose chronic disease such as asthma and genetic conditions. Suggestions for areas of innovation are listed with quotes in table 3.

## DISCUSSION

### Summary of main findings

There are areas of unmet need for POC tests in paediatric ambulatory care in England. Participants wanted more POC tests and technologies to be available. They thought they should be user-friendly and, where possible, fun. They felt that they could empower patients and their families when used at home, particularly in children with chronic disease. Clinicians wanted POC tests to give results that made a difference to clinical management, especially in the detection of acute serious illness in children for whom diagnosis is more challenging.

### Strengths and weaknesses of this study

Strengths of this study include the use of semistructured interviews, enabling an in-depth exploration of

the experiences of the heterogeneous participants with different backgrounds.[23] Purposive sampling with snowballing facilitated the interviewing of a broad range of clinicians and stakeholders on this topic. The participants had diverse job roles, work settings and levels of experience. This enabled a wide variety of perspectives to be captured including those from policymaking and industry. Important needs of particular groups of children were highlighted because specialist experts were purposively sampled.

However, the broad remit of the study meant that we were unable to cover every single test and paediatric clinical presentation, making 'data saturation'[20] difficult to achieve. Understanding of specific POC tests, as well as specific clinical presentations and contexts, could be examined in a more in-depth way in a focused study. Furthermore, although children's and parents' perspectives were mentioned by our participants, and some offered their own experiences as parents, their views were not specifically sought in this study. Finally, all participants were based in England. As such, our findings are applicable to English stakeholders and clinicians and may not be transferable to other settings.

### Findings in relation to other studies

Our finding of unmet needs corroborated one systematic meta-analysis which demonstrated that very few studies, limited to a handful of diseases, have shown benefit of POC tests in paediatric populations.[5] Concerns over lack of funding were similarly found in a survey of UK children's emergency departments and paediatric assessment units.[18] In keeping with the concerns expressed in that survey about quality assurance, our participants stated that they wanted tests with high specificity and sensitivity. In contrast to that survey, our participants did not express concern that there was lack of evidence surrounding the use of POC tests.

Our study also shared some findings with a qualitative systematic review assessing clinicians' attitudes towards POC blood tests in primary care settings in high-income countries.[14] For example, many of our participants thought that POC tests could facilitate early clinical decision-making, as did the clinicians in the systematic review. In our study, participants placed new importance on the use of POC tests and technologies for earlier detection of acute serious illness in children who present atypically, and for whom diagnosis is normally delayed as a result.

Our study highlighted that the convenient use of POC tests at home by patients and their families could bypass the need for clinician assessment and empower patients and families. This is in keeping with the NHS's promotion of Integrated Care Systems,[27] and development of better diagnostics to improve diagnostic bottlenecks and help tackle health inequalities.[28] Child health nurses have highlighted in an interview study that parents felt empowered by being able to take care of their child in a safe and structured way at home.[29] Our participants didn't express

the concern found in a Swedish study of hospital clinicians that POC testing at home may drive unnecessary testing.[19]

The preference of our participants for POC tests to be easy to use and avoid causing pain was also evident in a more focused interview study of English hospital clinicians.[17] Their belief that finger prick testing is acceptable has similarly been demonstrated in GP settings.[30] Our study highlighted new information that play, visualisation and reward are important components of successful POC tests and technologies in children.

Many of our participants wanted tests that would make a difference to clinical management—particularly to flag risk of serious clinical deterioration, and distinguish between viral and bacterial disease. This was also found by a qualitative study of English hospital healthcare workers.[17] Both that study and our study have raised the importance of particular pathogen testing for infection control—theirs RSV, ours SARS-CoV-2 and meningococcus. Many of our participants expressed a preference for panels of pathogens, as did the first study.

### Implications for clinicians, policymakers and industry

We found that UK clinicians and stakeholders were of the opinion that existing bedside tests were not fit-for-purpose in ambulatory care paediatrics. One priority should be refining and enhancing existing tests, for example, the measurement of oxygen saturations in young children.

Participants wanted POC tests to be routinely available in clinical practice with the potential for tests to be used by children and their carers at home. For diagnostic developers, our study offers evidence in favour of the design of POC tests and technologies that incorporate play and reward to make them more acceptable to children and their carers.

### Unanswered questions and future research

Further qualitative and health services research to evaluate preferred characteristics of POC tests and technologies from parents and children themselves is advised to guide future 'patient-up' development by industry. This study highlighted that this would be particularly important in children who present atypically, such as children with disability, and children diagnosed with cancer. This would enable more equitable representation of children with greater healthcare needs.

A variety of unmet needs for diagnostics in paediatric ambulatory care were identified by our study, such as reliable early detection of acute serious illness, and the 'holy grail' of differentiation between viral and bacterial illness. This provides support for investment in research and development in these areas.

**Acknowledgements** We acknowledge our participants who took the time to participate in our study.

**Contributors** MER: Interviews, project administration, data curation, formal analysis, writing—original draft, writing—review & editing, guarantor. CB: Writing—review & editing. OvH: Writing—review & editing. MG: Conceptualisation,

methodology, supervision, writing—review & editing. GH: Conceptualisation, funding acquisition, methodology, supervision, writing—review & editing.

**Funding** This research was funded by the National Institute for Health Research (NIHR) Community Healthcare MedTech and in Vitro Diagnostics Co-operative (CH MIC), award code MIC-2016-018. MER's salary was funded by an In-practice fellowship grant from the NIHR.

**Disclaimer** The views expressed are those of the authors and not necessarily those of the NHS, the NIHR or the Department of Health and Social Care.

**Competing interests** None declared.

**Patient and public involvement** Patients and/or the public were not involved in the design, or conduct, or reporting, or dissemination plans of this research.

**Patient consent for publication** Not applicable.

**Ethics approval** This study involves human participants and was approved by Medical Sciences Interdivisional Research Ethics Committee, University of Oxford, on 30 April 2019 (reference R63109/RE001); and LSHTM (London School of Hygiene & Tropical Medicine) MSc Research Ethics Committee on 14 May 2019 (reference 17436). Participants gave informed consent to participate in the study before taking part.

**Provenance and peer review** Not commissioned; externally peer reviewed.

**Data availability statement** Data are available upon reasonable request, subject to screening by a panel of the authorship team.

**ORCID iDs**
Meriel Elizabeth Raymond http://orcid.org/0000-0002-0442-9280
Gail Hayward http://orcid.org/0000-0003-0852-627X

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
