## [Reviewer comments · BMJ Open]

ARTICLE DETAILS

TITLE (PROVISIONAL)	Point-of-care Diagnostic Technology in Paediatric Ambulatory Care: a Qualitative Interview study of English Clinicians and Stakeholders
AUTHORS	Raymond, Meriel; Bird, Chris; van Hecke, Oliver; Glogowska, Margaret; Hayward, Gail

VERSION 1 – REVIEW

REVIEWER	Shabaninejad, Hosein University of Newcastle upon Tyne, Population Health Sciences Institute
REVIEW RETURNED	03-Dec-2021

GENERAL COMMENTS	This is a valuable research and its publication brings ideas for further research and sheds light on the area that could improve the use of POC tests in Health Care Systems. The only point that I could suggest is an overview of quantitative research in this area and if so, how the results of this research should be discussed in a line with the previous quantitative research.
---

REVIEWER	Rasti, Reza Karolinska Institute, Global Public Health
REVIEW RETURNED	23-Jan-2022

GENERAL COMMENTS	Manuscript bmjopen-2021-059103 Title: "Point-of-care diagnostic technology in paediatric ambulatory care: a qualitative interview study of clinicians and stakeholders" The aim of this study was to explore views and experiences from current use and need for POC diagnostics, of clinicians and stakeholders with interest in paediatric ambulatory care in the UK. The authors rightly state that the use of POC diagnostics in paediatrics is under-investigated. Thus, the rationale for the study is agreeable. However, the submitted manuscript has major issues that need to be addressed. In its current version, the manuscript seems incomplete.
--

I will start by pointing out its main deficiencies and move on to more specific comments.

First, the 'Discussion' section is one of the shortest I have encountered, and especially so for qualitative studies. Despite the shortage of studies on paediatric POCT use, there are a few recently published and highly relevant papers that has eluded mentioning by the authors.

Please see:

Rasti R, Brännström J, Mårtensson A, Zenk I, Gantelius J, Gaudenzi G, et al. Point-of-care testing in a high-income country paediatric emergency department: a qualitative study in Sweden. *BMJ open*. 2021;11(11):e054234.

Rasti R, Nanjebe D, Karlstrom J, Muchunguzi C, Mwanga-Amumpaire J, Gantelius J, et al. Health care workers' perceptions of point-of-care testing in a low-income country-A qualitative study in Southwestern Uganda. *PLoS One*. 2017;12(7):e0182005.

Pandey M, Lyttle MD, Cathie K, Munro A, Waterfield T, Roland D, et al. Point-of-care testing in Paediatric settings in the UK and Ireland: a cross-sectional study. *BMC Emergency Medicine*. 2022;22(1):6.

Li E, Dewez JE, Luu Q, Emonts M, Maconochie I, Nijman R, et al. Role of point-of-care tests in the management of febrile children: a qualitative study of hospital-based doctors and nurses in England. *BMJ Open*. 2021;11(5):e044510.

How do findings of the present study compare to those of the above stated publications? Which conclusions can be drawn? What new findings does the study present?

The present manuscript does not present any interpretation of its findings, let alone interpret or discuss similarities/differences to existing literature.

Secondly, *BMJ Open* has an international reach. Yet, this manuscript is highly UK focused in its shown interest as well as in the language used. It contains several jargon words/phrases, such as "snowballing", "ground up", "single sheet" and others, that are not established in the scientific community, and not readily apprehendable by most readers.

Also, the lack of a thorough description of the study setting and of the health system where the study is conducted makes it impossible for a non-UK reader to understand what a "Macmillan" GP, or "CCG" are. What is a "Foundation Year 1 doctor"?

There are numerous such examples where the chosen wording and labelling might be obvious to a clinician in the UK, but are not self-explanatory to anybody else.

	Title: It is unspecific with regards to where the study is conducted, and what is studied (perspectives of clinicians and stakeholders). Also, is it the technology that is studied, or the use of the technology? Abstract: The setting (UK) is not stated under 'Design, setting and participants' Results: Please consider starting with what the data analysis generated, i.e. "Data analysis yielded XX sub-themes and YY main themes, and then move on to describing what the (sub-)themes actually reflected. Also, the number of interviews (22) should be moved to the Result section, in line with the main body of the manuscript. Conclusions: What was the cause (benefits of POCT use) that made participants to support their use in paediatrics? What was it with some tests that made them non-fit for purpose? Could, or should they be refined, and why (for higher utility in paediatrics?). Strengths and limitations of this study: The first point claims in-depth exploration of experiences. The third point contradicts the first point by stating that depth was limited. Introduction: Page 3: Line 30: "to be less than 1%" needs reference. Line 34: ref marking (7) should be moved to end of sentence.
--	---

Line 37-38: Where was the increase in hospital admissions seen? In the UK? Elsewhere? Clarify. Also, do the authors speculate that the increase was due to diagnostic uncertainty? If so, needs a reference, and also clarification in order to open for POCTs to have potential of narrowing such diagnostic uncertainties.

Line 42: again, refrain from phrases such as “low numbers, high stakes”.

Line 45: here you present the chosen definition of POCTs, and from the definition it seems to specify in-vitro POCTs (“without needing to send a sample to a laboratory”). Yet, later in the manuscript you include other techniques, such as height measurement, etc. This does not comply with the presented definition.

Line 48-49: the claim needs references.

Line 49-52: please see the four articles stated above. Also, all articles describe more than the benefits of POCTs in paediatric, more importantly they illustrate disbenefits of POCTs and in how they are being used, or how they have been implemented.

Page 4:

Line 8: what are references 17 and 18? Does BMJ Open allow the use of such references, which cannot be found by readers of this manuscript?

Line 13: apps and wearables cannot be considered POC technologies with the chosen definition of such.

Line 15-16: Reference needed for this claim. Also, why can't paediatric needs be met with POCTs developed for use in adults? Explain to readers.

Line 19-20: Agree, paediatrics have higher use of POCTs. But this claim needs references, you can find such in the above articles.

Line 35: In this modern world “sex” would be a more correct term than “gender”.

Line 52-54: this could be understood as “saturation” was obtained. Yet, previously, in the Strengths/limitations section you stated that saturation was difficult to achieve. Contradictory.

Line 60: the entire last sentence (which continues into page 5) is unclear to me. Were questionnaires used during the interviews? Or did you use an interview/topic guide of sorts? The material used during the interviews need to be presented

Page 5:

Line 6-7: I do not agree that focus group discussions (FGDs) would be less likely to capture individual experiences. This claim needs justification and references.

FGDs would have been an excellent method, where the groups could have been modelled to be naturalistic, or according to profession. So the claim in the manuscript needs clarification and justification.

Line 13: where were the face-to-face interviews conducted?

Line 15-19: please see previous comment about presenting the topic guide. Also, what was the basis of the topic guide? Was it modelled on existing literature? If so, which literature? Or how were the questions chosen, on what grounds? Please clarify.

Also, please add a sentence about the duration of the interviews (e.g. interviews ranged 25 to 43 minutes, median 34 minutes), or something similar.

And which of the authors conducted the interviews? Who transcribed the audio recordings?

Line 35: write “six coded interviews ” instead of ‘6’

Line 44: Excellent that the participants were allowed to provide feedback on findings! But how were the findings presented to them, and how did they provide feedback?

Researcher characteristics: were any of the other authors/researchers more seasoned in qualitative research? Please describe.

Page 6:

Line 8-10: Here it is stated that only telephone interviews were conducted. Please move the last sentence to the 'Methods' section.

Line 38-43: the quote seems to be about POCUS. Although a good quote, again, the POCT definition presented earlier is not in line with the technology described by the quote.

Also, most established qualitative researchers advise against using any identifier for a quote. Hence, there is no need to detail the person behind a quote with a number, such as "Consultant clinician#6". Instead, "Consultant clinician" is sufficient.

Line 45-46, this sentence seems out of place as it is standing by itself without any corresponding quote, or anything else to refer to.

Line 48-52 holds an interesting view, and would benefit from a quote.

Line 55-57: how can spirometry, FeNO, and mental health questionnaires be considered as POC tests? Is a stethoscope a POCT? Are measurement tapes or waves POC tests, as they provide measurements of patients? In the manuscript, there doesn't seem to be any limits to what can be considered a POC test. Similarly, in the next section on page 7, how can measurers of vital signs, such as thermometers be POCTs?

Page 8:

	Line 13-15: “Participants reported that samples need to be ...” and later “This was particularly true for ... and blood pressure measurements”. A blood pressure measurement is not sampling. Line 22-29: “... technologies needed to be “fool proof” ... “. And later, this is exemplified by the description of how oxymeter devices are difficult to locate in the premises of health providers. I have difficulty in considering the displacement of equipment or devices to be anything else than handling issues instead of a sign of the device not being fool proof. So the pursuant quote, although a good one, is more an illustration of deficient implementation, instead of an illustration of how the oxymeter is not fool proof. Page 9, line 30-34: nice quote. Discussion: The entire discussion section needs finishing before it can be commented on. Finally, I apologize for the somewhat harsh comments. I believe studies such as this are highly needed and especially relevant in today’s world where the use of rapid antigen tests for COVID-19 have become widely used, not only in health care settings but also by the general public. Therefore, I am somewhat disappointed by this manuscript. The study itself seems to be ambitious, especially as you have succeeded in including stakeholders as their perspectives are in shortage in current literature. Also, you have a fair amount of participants, which should have generated data rich in content. However, the present manuscript falls short of delivering to its potential.
--	--

REVIEWER	Yadav, Kapil All India Institute of Medical Sciences, Centre for Community Medicine
REVIEW RETURNED	24-Jan-2022

GENERAL COMMENTS	Dear editor, Many thanks for sharing this manuscript with us. We have read them with great interest. The manuscript has been written around
---

	the use of point of care devices by clinicians for the pediatric population. However, we have a few submissions to make:  1. The manuscript misses out on the validity component of the POC devices- sensitivity, specificity, PPV, and NPV. The validity component will be required for the standardization usage of the devices. 2. The verification and monitoring of the analytical performance of the devices should include the Quality Control component, calibration verification, range verification, and method comparability. 3. Faster devices do not always lead to a better outcome. The POC devices would be useful in decreasing the waiting time for the result. However, the device's accuracy and precision will be good enough, is not discussed. 4. To promote the use of POC devices by clinicians, the emphasis for any laboratory devices should also be put on the scientific, economic, public health, political, and regulatory aspects. 5. The cost of these devices should be taken into consideration before making a decision. 6. Record keeping is necessary while using medical devices. With POC device usage by clinicians, maintenance of the devices will be very difficult. 7. In the manuscript, the negative responses from the clinicians are not discussed.
--	--

VERSION 1 – AUTHOR RESPONSE

Reviewer requests:

Reviewer: 1

Dr. Hosein Shabaninejad, University of Newcastle upon Tyne

Comments to the Author:

This is a valuable research and its publication brings ideas for further research and sheds light on the area that could improve the use of POC tests in Health Care Systems. The only point that I could suggest is an overview of quantitative research in this area and if so, how the results of this research should be discussed in a line with the previous quantitative research.

In the introduction we reference the only systematic review and meta-analysis we are aware of on the use of POC tests in paediatric ambulatory care globally. Many of the studies in the review were focussed on HIV and malaria in low-and-middle-income countries. Those in high-income countries focussed on respiratory tract infection and fever and predominantly studied POC C-reactive protein (CRP). In the UK POC CRP is not routinely used in children outside of a clinical trial setting.

We have expanded the summary of the quantitative research mentioned in the introduction accordingly:

A systematic review and meta-analysis of the clinical impact of POC tests in paediatric ambulatory care found few studies [5]. The use of malarial POC tests was found to reduce over-treatment by a third compared to usual care. HIV-POC tests improved early initiation of antiretroviral therapy compared to usual care. POC C-reactive protein may reduce immediate antibiotic prescribing for respiratory tract infections in low-and-middle-income countries, but evidence was lacking in high-income countries. The evaluation of POC tests for children often lags behind that for adults, for example with SARS-CoV-2 testing [16].

We have also added the following sentence to the discussion:

Our finding of unmet needs corroborated one systematic meta-analysis which demonstrated that very few studies, limited to a handful of diseases, have shown benefit of POC tests in paediatric populations [5].

Reviewer: 2

Dr. Reza Rasti, Karolinska Institute

Comments to the Author:

Manuscript bmjopen-2021-059103

Title: "Point-of-care diagnostic technology in paediatric ambulatory care: a qualitative interview study of clinicians and stakeholders"

The aim of this study was to explore views and experiences from current use and need for POC diagnostics, of clinicians and stakeholders with interest in paediatric ambulatory care in the UK. The authors rightly state that the use of POC diagnostics in paediatrics is underinvestigated. Thus, the rationale for the study is agreeable.

However, the submitted manuscript has major issues that need to be addressed. In its current version, the manuscript seems incomplete.

I will start by pointing out its main deficiencies and move on to more specific comments.

First, the 'Discussion' section is one of the shortest I have encountered, and especially so for qualitative studies.

The discussion has been re-worked on the suggestion of the editor- please see page 2 of this letter.

Despite the shortage of studies on paediatric POCT use, there are a few recently published and highly relevant papers that has eluded mentioning by the authors.

Please see:

1. Rasti R, Brännström J, Mårtensson A, Zenk I, Gantelius J, Gaudenzi G, et al. Point-of-care testing in a high-income country paediatric emergency department: a qualitative study in Sweden. *BMJ open*. 2021;11(11):e054234.

2. Rasti R, Nanjebe D, Karlstrom J, Muchunguzi C, Mwanga-Amumpaire J, Gantelius J, et al. Health care workers' perceptions of point-of-care testing in a low-income country-A qualitative study in Southwestern Uganda. *PLoS One*. 2017;12(7):e0182005.

3. Pandey M, Lyttle MD, Cathie K, Munro A, Waterfield T, Roland D, et al. Point-of-care testing in Paediatric settings in the UK and Ireland: a cross-sectional study. *BMC Emergency Medicine*. 2022;22(1):6.

4. Li E, Dewez JE, Luu Q, Emonts M, Maconochie I, Nijman R, et al. Role of point-of-care tests in the management of febrile children: a qualitative study of hospital-based

doctors and nurses in England. *BMJ Open*. 2021;11(5):e044510.

How do findings of the present study compare to those of the above stated publications?

Which conclusions can be drawn?

What new findings does the study present?

Thank you for drawing these papers to our attention, some of which were published at or just before the point of submission of our manuscript, but are valuable additions to the discussion. We have now mentioned 1., 3. and 4. in both the introduction and discussion sections which have strengthened these. We did not include reference to 2. as this study does not have a focus on paediatric POC tests, but rather all patients. It is also focussed in a low-income setting, which has less relevance to our high-income study based in England.

4. had previously been referenced in our manuscript with the references listed below. We had not been aware that the results had been formally published as the authors had not responded to an email from us asking whether it had. These have now been updated.

Candidate 111003, Dewez M, Yeung S. MSc Project Report The Perceptions of English Paediatricians on the Use of Point-of-Care-Tests (POCTs) in Assessing Febrile Children . 2018;(September):1–50.

Candidate 110875, Dewez M, Yeung S. MSc Project Report Perceptions of English Emergency Department Healthcare Providers on the Use of Rapid Diagnostic Tests (RDTs) in Febrile Children: A Qualitative Study. 2018;(September):1–88.

The following page numbers refer to the word document Point-of-care Diagnostic Technology in Paediatric Ambulatory Care A Qualitative Interview Study of Clinicians and Stakeholders_1.5_Main Document_clean.docx.

The present manuscript does not present any interpretation of its findings, let alone interpret or discuss similarities/differences to existing literature.

We disagree with this statement “the present manuscript does not present any interpretation of its findings”. An example of interpretation is given below:

Discussion, Findings in relation to other studies, p11:

Our study highlighted new information that play, visualisation and reward are important components of successful POC tests and technologies in children.

Discussion, Implications for clinicians, policymakers, and industry p11:

For diagnostic developers, our study offers evidence in favour of the design of POC tests and technologies that incorporate play and reward to make them more acceptable to children and their carers

Secondly, *BMJ Open* has an international reach. Yet, this manuscript is highly UK focused in its shown interest as well as in the language used. It contains several jargon words/phrases, such as “snowballing”, “ground up”, “single sheet” and others, that are not established in the scientific community, and not readily apprehendable by most readers.

- The literature suggests “Snowball sampling is a commonly employed sampling method in qualitative research, used in medical science and in various social sciences, including sociology, political science, anthropology and human geography” (<https://www.ncbi.nlm.nih.gov/pmc/articles/PMC6104950/#>).
- The mention of a “ground up” approach is referenced.
- We believe “single sheet” to be self-explanatory.

Also, the lack of a thorough description of the study setting and of the health system where the study is conducted makes it impossible for a non-UK reader to understand what a “Macmillan” GP, or “CCG” are. What is a “Foundation Year 1 doctor”?

- We have changed

A Macmillan GP with palliative care as a specialist interest

To:

A Macmillan GP (GP with palliative care as a specialist interest)

- We have added:

CCGs (Clinical Commissioning Groups; groups of general practices which come together in each area to commission services for their patients and population)

- We’ve changed reference “Foundation Year 1 doctor” to “Foundation Year 1 Doctor (junior doctor in their first year of practice)” to make their role clearer.

There are numerous such examples where the chosen wording and labelling might be obvious to a clinician in the UK, but are not self-explanatory to anybody else.

There are no further examples given here to address, and our review of the manuscript does not suggest any others.

Title: It is unspecific with regards to where the study is conducted, and what is studied (perspectives of clinicians and stakeholders). Also, is it the technology that is studied, or the use of the technology?

- We have added “English” to the title.
- We believe it’s clear that perspectives are explored given that it is a qualitative study
- Both technology itself and its use are explored, and we feel this is covered by the title.

Abstract:

The setting (UK) is not stated under ‘Design, setting and participants’

“in England” has been added to the first paragraph of the abstract.

Results: Please consider starting with what the data analysis generated, i.e. “Data analysis yielded XX sub-themes and YY main themes, and then move on to describing what the (sub-)themes actually reflected.

Unfortunately, there is insufficient word count in the abstract to add the main themes. These can be found in the main text.

Also, the number of interviews (22) should be moved to the Result section, in line with the main body of the manuscript.

We have adjusted this.

Conclusions: What was the cause (benefits of POCT use) that made participants to support their use in paediatrics? What was it with some tests that made them non-fit for purpose? Could, or should they be refined, and why (for higher utility in paediatrics?).

Again, this cannot be expanded due to the limit of the word count. This is explored in the discussion.

Strengths and limitations of this study:

The first point claims in-depth exploration of experiences. The third point contradicts the first

point by stating that depth was limited.

We have changed:

However, the breadth of the study limited the depth to which we could explore any specific clinical presentations or contexts and made “data saturation” (18) difficult to achieve.

To:

However, the broad remit of the study meant that we were unable to cover every single test and paediatric clinical presentation, making “data saturation” (18) difficult to achieve.

Introduction:

Page 3:

Line 30: “to be less than 1%” needs reference.

Done

Line 34: ref marking (7) should be moved to end of sentence.

Done

Line 37-38: Where was the increase in hospital admissions seen? In the UK? Elsewhere? Clarify.

“in England” added

Also, do the authors speculate that the increase was due to diagnostic uncertainty? If so, needs a reference, and also clarification in order to open for POCTs to have potential of narrowing such diagnostic uncertainties.

We highlight that diagnostic uncertainty, and that short stay avoidable admissions have increased. We do not state here that these are linked.

Line 42: again, refrain from phrases such as “low numbers, high stakes”.

This is a quote, not our own phrase. We feel it illustrates the issue well.

Line 45: here you present the chosen definition of POCTs, and from the definition it seems to specify in-vitro POCTs (“without needing to send a sample to a laboratory”). Yet, later in the manuscript you include other techniques, such as height measurement, etc. This does not comply with the presented definition.

To clarify this we have added the following:

- to introduction (p3):

POC (point-of-care) tests can be defined as any test performed near a patient or clinic with results available during a clinical visit [14,15]. Point-of-care technology includes measurements taken at the bedside, such as smartphone applications and wearables.

- to Methods, Interviews (p5):

Participants were informed “by point-of-care tests and technologies, we mean any diagnostic technology to include tests on bodily fluids, imaging, wearables, digital technology, and smart phone apps”.

Line 48-49: the claim needs references.

Done

Line 49-52: please see the four articles stated above. Also, all articles describe more than the benefits of POCTs in paediatric, more importantly they illustrate disbenefits of POCTs and in how they are being used, or how they have been implemented

We have added mention of the Li reference into the manuscript, see page 4-5 of this letter. Findings of disbenefits of POCTs in the literature have now been expanded in the introduction.

Page 4:

Line 8: what are references 17 and 18? Does BMJ Open allow the use of such references, which cannot be found by readers of this manuscript?

These two references mentioned above are the following:

Candidate 111003, Dewez M, Yeung S. MSc Project Report The Perceptions of English Paediatricians on the Use of Point-of-Care-Tests (POCTs) in Assessing Febrile Children . 2018;(September):1–50.

Candidate 110875, Dewez M, Yeung S. MSc Project Report Perceptions of English Emergency Department Healthcare Providers on the Use of Rapid Diagnostic Tests (RDTs) in Febrile Children: A Qualitative Study. 2018;(September):1–88.

They have been replaced with:

Li E, Dewez JE, Luu Q, Emonts M, Maconochie I, Nijman R, et al. Role of point-of-care tests in the management of febrile children: a qualitative study of hospital-based doctors and nurses in England. *BMJ Open*. 2021;11(5):e044510.

Line 13: apps and wearables cannot be considered POC technologies with the chosen definition of such.

As discussed above, we have clarified the scope of the research further in the introduction.

Introduction: (p3):

Point-of-care technology includes measurements taken at the bedside, such as smartphone applications and wearables.

Line 15-16: Reference needed for this claim.

Also, why can't paediatric needs be met with POCTs developed for use in adults? Explain to readers.

We do not believe a reference is needed for the following statement:

The diagnostic needs in paediatric ambulatory care are unlikely to be met by diagnostics which have been developed with an adult population primarily in mind.

We have already stated (introduction, p3):

Children present with a different disease spectrum to adults, having a high incidence of acute infections [3].

Furthermore, children have the potential to deteriorate more quickly than adults [4].

We have added (introduction, p4):

Children are not “mini adults” and have specific needs that should be addressed in order for diagnostics to be helpful in a clinical setting. These might include the requirement for rapid diagnosis, smaller sample volumes and less invasive procedures.

Line 19-20: Agree, paediatrics have higher use of POCTs. But this claim needs references,

you can find such in the above articles.

We do not think the following requires a reference:

In order to stimulate the development and evaluation of POC diagnostic technology which is of greatest benefit in paediatric healthcare it is important to understand the current experience of those using these technologies and identify areas of unmet need.

Line 35: In this modern world “sex” would be a more correct term than “gender”.

Participants were not asked about biological sex, and therefore gender is the more appropriate term here.

Line 52-54: this could be understood as “saturation” was obtained. Yet, previously, in the Strengths/limitations section you stated that saturation was difficult to achieve. Contradictory.

We have changed the following:

The decision to stop interviewing, when little new information was emerging and there was sufficient explanation for the emerging themes, was discussed and agreed among the research team.

To:

The decision to stop interviewing, when sufficient information had emerged and there was satisfactory explanation for the emerging themes, was discussed and agreed among the research team.

Line 60: the entire last sentence (which continues into page 5) is unclear to me. Were questionnaires used during the interviews? Or did you use an interview/topic guide of sorts? The material used during the interviews need to be presented
Were questionnaires used during the interviews? Or did you use an interview/topic guide of sorts?

We assume you’re referring to the following sentence:

Interviews were chosen in preference to questionnaires to enable in-depth exploration of the experiences of the heterogeneous participants (21) , through interviewer and interviewee interaction, and exploration of details which were significant to either party as the interview progressed.

We have reviewed this section and edited to make sure it was clear that our method was qualitative interviews only

Draft topic guides for the interviews with clinicians and stakeholders were developed to address the study objectives

Page 5:

Line 6-7: I do not agree that focus group discussions (FGDs) would be less likely to capture individual experiences. This claim needs justification and references.

FGDs would have been an excellent method, where the groups could have been modelled to be naturalistic, or according to profession. So the claim in the manuscript needs clarification and justification.

The decision made was justified in the manuscript as follows, p4. None of the sentences below require references.

A focus group discussion of a wide range of professionals would be less likely to capture these individual experiences. Focus-group discussion was also avoided due to logistical difficulty in

arranging group clinician sessions; need for HRA (Health Research Authority) approval for interviews occurring on NHS (National Health Service) premises; and divergence of stakeholder interests.

Line 13: where were the face-to-face interviews conducted?

This information is available in the results section, p5:

Due to participant preference and the COVID-19 pandemic, all interviews were conducted by telephone.

As per page 12 of this letter, this has been moved to the methods section.

Line 15-19: please see previous comment about presenting the topic guide. Also, what was the basis of the topic guide? Was it modelled on existing literature? If so, which literature? Or how were the questions chosen, on what grounds? Please clarify.

The draft topic guide has now been included in Supplementary Materials.

We have added the following, p5:

These were based on available literature, and drew on issues from topic guides for other studies we have conducted around clinicians' views of POC testing [24,25].

Also, please add a sentence about the duration of the interviews (e.g. interviews ranged 25 to 43 minutes, median 34 minutes), or something similar.

We have added to results, p5:

The interviews lasted an average of 35 minutes.

And which of the authors conducted the interviews?

This is already explained in the manuscript under:

Methods, Interview, p4:

Qualitative semi-structured individual interviews were conducted by the primary researcher MR.

Author contributions, p12:

MR- Interviews

Who transcribed the audio recordings?

We have added:

by a single professional transcriber.

To:

Methods, interview, p4:

Interviews were recorded using a digital audio-recorder and transcribed verbatim

Line 35: write "six coded interviews " instead of '6'

Done

Line 44: Excellent that the participants were allowed to provide feedback on findings! But how were the findings presented to them, and how did they provide feedback?

We have changed (Methods, Analysis, p5):

Participants provided feedback on the findings.

To the following:

Participants were provided with the results section and given two weeks to provide feedback.

Researcher characteristics: were any of the other authors/researchers more seasoned in qualitative research? Please describe.

The following has been added to Researcher characteristics, p5:

MG is a specialist qualitative researcher.

Page 6:

Line 8-10: Here it is stated that only telephone interviews were conducted. Please move the last sentence to the 'Methods' section.

Done

Line 38-43: the quote seems to be about POCUS. Although a good quote, again, the POCT definition presented earlier is not in line with the technology described by the quote.

As before, the following has been added to introduction (p4):

Point-of-care technology includes measurements taken at the bedside, such as smartphone applications and wearables.

Also, most established qualitative researchers advise against using any identifier for a quote. Hence, there is no need to detail the person behind a quote with a number, such as "Consultant clinician#6". Instead, "Consultant clinician" is sufficient.

Using an interview number denotes where a quotation comes from to demonstrate transparency and confirmability, linking the raw data in the interviews to emergent themes.

Giving the designation provides an immediate context around the quotation, and has been a standard way in how we have presented the qualitative research we have successfully published in a variety of journals (<https://doi.org/10.1186/s12875-021-01571-0>, <https://doi.org/10.1186/s12888-021-03067-8>, <https://doi.org/10.1186/s12875-020-01316-5>, <https://doi.org/10.1371/journal.pone.0228687>).

Line 45-46, this sentence seems out of place as it is standing by itself without any corresponding quote, or anything else to refer to.

This refers to the following sentence, p6:

Delayed laboratory results would be more likely to be interpreted by a clinician who had not seen the child.

We have moved this to the end of paragraph 1, under 1a: POC tests facilitate early decision-making, p6.

Line 48-52 holds an interesting view, and would benefit from a quote.

We have added the following quote, p6:

"they'd been back and forwards to the GP with tiredness or a bit of a viral infection... and it was only when they got into A&E [Accident and Emergency]... that the blood tests [were] done and the leukaemia was found... probably a barrier for us in primary [care] at the moment is that we would have to refer the patient to... the hospital... but if we could just do it in primary care that probably would... transform that sort of diagnosis". [Macmillan GP, Clinician#5]

Line 55-57: how can spirometry, FeNO, and mental health questionnaires be considered as

POC tests? Is a stethoscope a POCT? Are measurement tapes or waves POC tests, as they provide measurements of patients? In the manuscript, there doesn't seem to be any limits to what can be considered a POC test. Similarly, in the next section on page 7, how can measurers of vital signs, such as thermometers be POCTs?

As before, the following has been added to introduction (p4):

Point-of-care technology includes measurements taken at the bedside, such as smartphone applications and wearables

This could cover novel ways of taking measurements, for example with new devices or smartphones. A stethoscope wouldn't be a POC test, but a stethoscope wired to a smartphone device that might aid its interpretation could be a POC technology.

Page 8:

Line 13-15: "Participants reported that samples need to be ..." and later "This was particularly true for ... and blood pressure measurements". A blood pressure measurement is not sampling.

We have changed, p8:

Participants reported that samples need to be easy to obtain

To:

Participants reported that POC tests need to be easy to perform

Line 22-29: "... technologies needed to be "fool proof" ... ". And later, this is exemplified by the description of how oxymeter devices are difficult to locate in the premises of health providers. I have difficulty in considering the displacement of equipment or devices to be anything else than handling issues instead of a sign of the device not being fool proof. So the pursuant quote, although a good one, is more an illustration of deficient implementation, instead of an illustration of how the oxymeter is not fool proof.

The "fool proof" quotation is an independent quotation by Emergency Department Consultant Clinician#6.

The quote "With younger kids..." exemplifies the sub-theme "End-users should find POC tests quick and easy to use", and other points in the paragraph above it, such as "Participants reported that where tests were not easy to use, it put them off using them".

Page 9, line 30-34: nice quote.

Thank you.

Discussion:

The entire discussion section needs finishing before it can be commented on.

Finally, I apologize for the somewhat harsh comments. I believe studies such as this are highly needed and especially relevant in today's world where the use of rapid antigen tests for COVID-19 have become widely used, not only in health care settings but also by the general public. Therefore, I am somewhat disappointed by this manuscript. The study itself seems to be ambitious, especially as you have succeeded in including stakeholders as their perspectives are in shortage in current literature. Also, you have a fair amount of participants, which should have generated data rich in content. However, the present manuscript falls short of delivering to its potential.

Thanks for your helpful and detailed review – we believe that responding to your comments has strengthened the manuscript.

Dear editor,

Many thanks for sharing this manuscript with us. We have read them with great interest. The manuscript has been written around the use of point of care devices by clinicians for the pediatric population.

However, we have a few submissions to make:

1. The manuscript misses out on the validity component of the POC devices- sensitivity, specificity, PPV, and NPV. The validity component will be required for the standardization usage of the devices.
2. The verification and monitoring of the analytical performance of the devices should include the Quality Control component, calibration verification, range verification, and method comparability.
3. Faster devices do not always lead to a better outcome. The POC devices would be useful in decreasing the waiting time for the result. However, the device's accuracy and precision will be good enough, is not discussed.
4. To promote the use of POC devices by clinicians, the emphasis for any laboratory devices should also be put on the scientific, economic, public health, political, and regulatory aspects.
5. The cost of these devices should be taken into consideration before making a decision.
6. Record keeping is necessary while using medical devices. With POC device usage by clinicians, maintenance of the devices will be very difficult.
7. In the manuscript, the negative responses from the clinicians are not discussed.

The reviewers raise interesting points about features of devices which may be of value to consider. However, this was an exploratory qualitative study to look at existing experience and perceptions of POC tests with children. This means that the features of value to clinicians were not pre-specified by the research team but were identified through the process of interviewing clinicians and stakeholders. The focus of the qualitative study was to explore frontline clinicians' and stakeholders' experiences of using POC diagnostic technology in children in the UK rather than the specific technical requirements and detailed characteristics of every POC test.

In fact, a number of the issues the reviewer feels might be relevant were mentioned by our interviewees.

For instance, high sensitivity and specificity were mentioned by our participants as being generally a desirable feature of a successful POC diagnostic. We have added a mention of sensitivity and specificity under "2d: POC tests should make a difference to clinical management", page 9, as follows:

Many of them expressed a wish for tests with "good sensitivity and specificity to be reliable" [Foundation Year 1 Doctor, Clinician#8].

The following issues were already mentioned in the manuscript, as per the examples below:

- Economic context, or cost:

They also thought that cost, for example of FeNO and peripheral oxygen saturation monitors, could limit accessibility and lead to "inequitable distribution" [Asthma nurse Clinician#4] (page 7)

- Public health:

The importance of exact pathogen detection in the context of public health was also raised, with implications for contact-tracing and vaccination when meningococci and SARS-CoV-2 were detected. (page 9)

This is in keeping with the NHS's promotion of Integrated Care Systems (page 11)

- Lack of negative views of POC tests:

Many challenges of POC tests are explored in the section: Theme 2: Areas for improvement for POC tests and technologies. The following taken from the manuscript demonstrates this:

"need to be easy to obtain to avoid causing pain and stress" (page 8)

"Many participants stated that urine samples (see Table 3), peak flows and spirometry could be challenging for younger children to perform" (page 8)

“Participants reported that where tests were not easy to use, it put them off using them. They frequently gave the example of measuring peripheral oxygen saturations which posed a logistical challenge in primary care as it was often difficult to locate equipment and obtain a reliable result” (page 8)

VERSION 2 – REVIEW

REVIEWER	Rasti, Reza Karolinska Institute, Global Public Health
REVIEW RETURNED	25-Apr-2022

GENERAL COMMENTS	Thank you for the revised manuscript. It now reads better than initially. I have no additional comments related to the content, only a few minor points:  1. The referenced pages in the SRQR need to be corrected. E.g. Context is on page 5, as is Sampling strategy. Additional corrections might be needed. 2. There are different font sizes used. but I guess this will be checked by the editorial board. 3. The funding source doesn't need to be included in the Abstract. Removing the sentence would trim the word count of the Abstract. Best of luck!
---

REVIEWER	Yadav, Kapil All India Institute of Medical Sciences, Centre for Community Medicine
REVIEW RETURNED	21-Mar-2022

GENERAL COMMENTS	We read with great interest the revised version of the manuscript submitted by authors. We recommend the same for publication with no further changes suggested from our end.
---